# Optimizing Lane Departure Warning System towards AI-Centered Autonomous Vehicles

**DOI:** 10.3390/s24082505

**Published:** 2024-04-13

**Authors:** Siwoo Jeong, Jonghyeon Ko, Sukki Lee, Jihoon Kang, Yeni Kim, Soon Yong Park, Sungchul Mun

**Affiliations:** 1Department of Sports Rehabilitation Medicine, Kyungil University, Gyeongsan 38428, Republic of Korea; swj736@kiu.ac.kr; 2Convergence Institute of Human Data Technology, Jeonju University, Jeonju 55069, Republic of Korea; whd1gus2@jj.ac.kr; 3Korea Institute of Civil Engineering and Building Technology, Goyang 10223, Republic of Korea; oksk@kict.re.kr; 4Department of Electronic Engineering, Inha University, Incheon 22201, Republic of Korea; jhkang57@inha.ac.kr; 5Department of Industrial Engineering, Jeonju University, Jeonju 55069, Republic of Korea; kimyeni8757@jj.ac.kr; 6Division of Urban Transportation Research, Seoul Institute, Seoul 06756, Republic of Korea; 7Department of Data Engineering, Jeonju University, Jeonju 55069, Republic of Korea

**Keywords:** LDWS, autonomous vehicle, road markings, retro-reflectivity, environmental conditions, simulation framework

## Abstract

The operational efficacy of lane departure warning systems (LDWS) in autonomous vehicles is critically influenced by the retro-reflectivity of road markings, which varies with environmental wear and weather conditions. This study investigated how changes in road marking retro-reflectivity, due to factors such as weather and physical wear, impact the performance of LDWS. The study was conducted at the Yeoncheon SOC Demonstration Research Center, where various weather scenarios, including rainfall and transitions between day and night lighting, were simulated. We applied controlled wear to white, yellow, and blue road markings and measured their retro-reflectivity at multiple stages of degradation. Our methods included rigorous testing of the LDWS’s recognition rates under these diverse environmental conditions. Our results showed that higher retro-reflectivity levels significantly improve the detection capability of LDWS, particularly in adverse weather conditions. Additionally, the study led to the development of a simulation framework for analyzing the cost-effectiveness of road marking maintenance strategies. This framework aims to align maintenance costs with the safety requirements of autonomous vehicles. The findings highlight the need for revising current road marking guidelines to accommodate the advanced sensor-based needs of autonomous driving systems. By enhancing retro-reflectivity standards, the study suggests a path towards optimizing road safety in the age of autonomous vehicles.

## 1. Introduction

The rapid evolution of autonomous vehicle technology has brought the need for precise lane recognition capabilities, which are crucial for vehicle safety and operational efficiency, into sharp focus [1,2,3,4]. These advancements necessitate sophisticated systems capable of accurately detecting and interpreting road markings, which are fundamental to vehicular navigation and control. In this context, the development and refinement of technologies like the lane departure warning system (LDWS) have become increasingly significant [5,6,7,8]. This progression is not just a technological evolution but also a safety imperative, as accurate lane recognition directly correlates with reducing road accidents and enhancing traffic management. The integration of such technologies is a critical component in advancing the capabilities of automated driving, fundamentally altering conventional approaches to road safety and vehicle control mechanisms.

LDWS, a key component in advancing driver assistance technologies, actively contributes to road safety by alerting drivers of unintentional lane deviations [6,7,8,9,10]. The performance of LDWS is dependent on the clarity and detectability of road markings, which can be influenced by factors such as weather, wear, and lighting conditions [11,12,13]. That is, this system’s functionality is intricately linked to environmental factors, particularly road marking visibility and conditions. The dependence emphasizes the necessity of understanding and adapting to the environmental variables that impact LDWS efficiency in autonomous vehicles, as maintaining its consistent functionality is important for road safety [14,15,16,17].

Ensuring high retro-reflectivity in road markings is crucial for maintaining consistent functionality of LDWS in autonomous vehicles [18], especially under varying weather conditions. The determination of a specific retro-reflectivity threshold that reliably supports lane recognition by LDWS is vital for vehicle safety. Despite its importance, there exists a research gap in identifying a reliable retro-reflectivity threshold from the perspective of LDWS. Few studies have addressed this issue [19], pointing to the need for further research focused on retro-reflectivity thresholds in relation to LDWS. Moreover, current domestic standards in South Korea for road markings are predominantly designed with general drivers in mind, primarily focusing on installation, maintenance, and repainting guidelines (Table 1). These standards have not yet incorporated specific criteria for LDWS, which rely on machine perception of road boundaries for navigation. As a result, the existing guidelines, which only consider retro-reflectivity values suitable for general driving conditions, may not suffice for autonomous driving systems. This highlights a critical need for revised guidelines that ensure clear and consistent road marking interpretations under diverse operational conditions, thereby closing the safety gaps in autonomous vehicle navigation.

The objective of this research was to establish a threshold of retro-reflectivity for road markings that vision-based LDWS can reliably detect, thus proposing a set of reflective performance guidelines. This research represents one of the approaches to quantitatively defining specific retro-reflectivity thresholds for road markings, a critical aspect which has previously been underexplored, especially from the LDWS perspective, in autonomous vehicles. The experiments were designed to test the lane recognition rate of LDWS across a spectrum of retro-reflectivity levels for white, yellow, and blue road lines. We systematically combined various conditions, including different levels of rainfall, day and night cycles, and vehicle light statuses (on/off), to comprehensively evaluate the performance of LDWS under realistic and controlled conditions. This approach allowed us to determine the critical retro-reflectivity values necessary for the LDWS to function effectively and safely, regardless of environmental conditions. Building upon these findings, we developed a simulation framework that utilizes established thresholds to evaluate road marking maintenance strategies. In particular, as the cost aspect of the maintenance strategies is a crucial objective for road management authorities [20,21,22], this framework was designed to optimize the balance between cost-effectiveness and the stringent safety requirements of autonomous vehicle ecosystems.

## 2. Materials and Methods

### 2.1. LDWS Functionality Tests

The LDWS functionality tests were conducted in strict compliance with the safety standards for autonomous vehicles set by the Ministry of Land, Infrastructure, and Transport. The vehicle test conditions meticulously maintained the ambient temperature on the test track within the range of 0 to 45 degrees Celsius. The procedure involved a vehicle equipped with LDWS using the front-view camera (QX1000, Mando Co., Ltd., Seoul, Republic of Korea), which accelerated from a stationary state to 60 km/h. This camera system was tasked with capturing front-view imagery, while lane detection was conducted using MobileEye’s eyeQ5 processor, specifically designed for advanced image recognition capabilities in LDWS applications. This configuration is widely used in LDWS vehicles operating in Korea, ensuring the relevance and applicability of our test results. The tests included executing a lane departure at a specified lateral speed within a controlled weather simulation section (Figure 1). The effectiveness of the LDWS was primarily assessed based on the road marking recognition rate. The road marking recognition rate (%, accuracy) was calculated as a percentage of correct recognitions, determined by dividing the number of road markings recognized by the total number of trials, then multiplying by 100%. To ensure the robustness of our data, each condition within the experiment underwent ten trials, allowing for a comprehensive assessment of the LDWS’s performance under various environmental scenarios.

### 2.2. Experimental Scenarios

The experimental scenarios were designed to assess the luminance of road markings at various stages of road marking wear, specifically for white, yellow, and blue markings. Prior to field application, a preliminary analysis was conducted to determine the rate of luminance reduction due to surface abrasion, ensuring that the scenarios were reflective of actual road conditions. To rigorously evaluate the LDWS’s capability to detect road markings under various degrees of wear and environmental conditions, the scenarios integrated factors such as color, wear extent (considering the degree of erosion), luminance, and weather conditions (rainfall). To reflect the complexities of realistic driving environments, the testing conditions were varied. Rainfall intensities were set at incremental levels of dry road surface, wet road surface, 20 mm/h, and 40 mm/h to simulate wet conditions. In terms of road marking wear, the experiments included stages of various degradations. The luminance conditions were controlled to simulate the visibility during the day and at night, with specific scenarios including daytime brightness, nighttime with street lighting and vehicle headlight, and nighttime with only vehicle headlights on (Table 2). These detailed parameters, in conjunction with a controlled track environment that replicated actual road conditions, allowed for a series of repeated trials to provide a thorough assessment of the LDWS’s performance across a spectrum of realistic and challenging driving scenarios.

### 2.3. Scenario Implementation

The Yeoncheon SOC (System of Control) Demonstration Research Center in Korea was utilized to establish diverse environmental conditions for analytical evaluation. The total length of the weather simulation test section spanned 200 m. The initial 100 m, located outside the tunnel shield, facilitated vehicle acceleration (Figure 1). The subsequent 100 m inside the tunnel shield were dedicated to performing lane changes and calculating the lane detection rate. Within the shielded section, precipitation was synthetically generated, ranging from 20 mm/h to 40 mm/h (Figure 2). Lighting conditions were regulated according to the time of day and the use of tunnel illumination. Daytime tests ran from 12:00 to 15:00, and nighttime sessions were performed after 21:00 with adjustable tunnel lighting (Table 2).

### 2.4. Vehicle’s Speed and Trajectory

The methodology for evaluating the LDWS’s detection accuracy was developed. The LDWS-equipped vehicle sufficiently accelerated outside of a shielded area and then passed through a simulation zone (tunnel shield) to execute a lane change within a specified section. It enables the measurement of the system’s recognition rate under controlled yet realistic conditions. The vehicle’s speed was maintained at 60 km/h, the operational speed for the LDWS, ensuring the system’s activation during testing. The lateral departure angle and speed were calculated to result in a trajectory with a tangent of 3.5/40 (equivalent to an angle of 5.01 degrees), corresponding to a lateral speed of 1.458 m/s (Figure 1). The trials were designed to minimize error and provide an accurate assessment of the LDWS’s performance in various weather-induced visibility conditions.

### 2.5. Preliminary Test

Prior to the main experiments of the LDWS’s capability tests, a series of preliminary experiments was conducted at the Yeoncheon SOC Demonstration Research Center to determine the relationship between the number of grindings and retro-reflectivity reduction. The test track was marked with white, yellow, and blue lane markings (Figure 3A). These markings underwent a systematic abrasion process using a grinder (Figure 3B), followed by measurements of the retro-reflective coefficient. Measurement points were marked alongside the markings. Subsequent to grinding, these points underwent multiple assessments to ensure the reliability of the data. The retro-reflective coefficients were obtained using a retro-reflectometer (LTL-XL, Delta Corp Holdings Ltd., Hørsholm, Denmark), which was calibrated before each measurement session with standard samples provided by the manufacturer (Figure 3C).

The preliminary findings indicated a decline in the retro-reflective coefficient with successive grinding frequency for test track lane markings (Figure 4). The results confirmed the SOC center’s capability to effectively simulate road marking wear and retro-reflectivity degradation.

### 2.6. Experimental Procedures

The LDWS road marking recognition rate experiment followed a protocol that incorporated varying weather conditions. The procedure included an initial application of road markings, simulation of wear through repeated grinding (Table 3), verification of retro-reflective coefficients, and a sequence of tests in both daytime and nighttime conditions, replicated ten times each to maintain experimental consistency. Additional tests replicating rainy conditions were also repeated ten times to ensure robust data.

### 2.7. Simulation Framework for Efficient Road Management

To enhance the practical applications of our experimental findings, a simulation framework was developed to analyze the cost-effectiveness of various road marking repainting scenarios. This framework specifically evaluated the impact of repainting on the performance of LDWS, particularly under challenging weather conditions. Central to this framework was the monitoring of road markings’ retro-reflectivity, assessing whether it fell below the thresholds outlined in Table 1. As seen in Figure 4, the retro-reflectivity of road markings tends to be non-linearly degraded by the frequency of grindings. This follows existing studies [23,24,25] where the retro-reflectivity of road lane markings in practice is degraded with a non-linear shape by time and the number of passing vehicles. In this regard, a non-linear regression model with a power regression design ($y=\beta_{0}+\beta_{1}\cdot x^{\rho }$) was employed to establish the LDWS recognition rate and the retro-reflectivity of road markings. This model took into account variations in luminance and weather conditions, where x denoted the target recognition rate and *y* represented the required retro-reflectivity. The model’s hyperparameter, ρ, which indicates the change in the retro-reflectivity with respect to the target recognition rate, was optimized using a grid search method within the range of [0, 3], aiming to maximize the *r*^2^ statistics and thus providing a more precise estimation of the necessary retro-reflectivity thresholds (Figure 5). The non-linear model bridged the gap between theoretical data and actionable guidelines.

In its practical application, the framework facilitated cost comparisons among different road marking paint products. Key factors considered in this comparison included: (1) the repainting cost per kilometer for each product (*C*); (2) the initial retro-reflectivity of these products (*W*); and (3) the duration taken for each product’s retro-reflectivity to diminish by one level of wear (*D*), a measure that varies based on regional environmental conditions. For this simulation, three hypothetical products were assumed (Table 4): *A*_1_ (currently used in our research), *A*_2_ (a more expensive but higher quality alternative to *A*_1_), and *A*_3_ (a cheaper but lower quality option compared to *A*_1_). The simulation framework, denoted as $SIM:\left(TR,ST,LW,C,W,D\right)\to Cost$ (Algorithm 1), calculated the total cost of maintaining a specified target recognition rate (*TR*) over a certain simulation period (*ST*) under various luminance and weather conditions (*LW*). This simulation aided in identifying the most efficient repainting strategy, balancing cost, and LDWS performance.
**Algorithm 1: Simulation Framework to Calculate Repainting Cost**   Input: *TR*, *ST*, *LW*, *C*, *W*, *D*   Output: TotalCost**1** Update of variables: assign *D* to Table 3**2** NewThreshold〈white, yellow, blue〉 ← predict retro-reflectivity by the fitted non-linear model based on Table 5, Table 6 and Table 7 with inputs of *TR*, *LW*
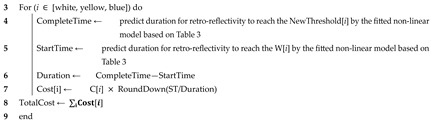


## 3. Results

### 3.1. White Lane

In the replicated weather simulation experiments for white lane markings, a 100% recognition rate was achieved during the day across all rainfall conditions, including 20 and 40 mm/h. However, nighttime conditions revealed that variations in lane recognition rates correlated with decreases in the retro-reflective coefficient. Repeated trials on dry and wet surfaces at night demonstrated changes in recognition rates. Regardless of nighttime lighting, a recognition rate above 90% was consistently observed, but at lower retro-reflectivity values of 142 mcd/(m^2^∙Lux) and 104 mcd/(m^2^∙Lux), the recognition rates dropped to 90% (Table 5).

Under nocturnal rainfall conditions, recognition rates fell below 70% for retro-reflective values of 142 mcd/(m^2^∙Lux) and 104 mcd/(m^2^∙Lux). Specifically, at a rainfall intensity of 20 mm/h with a retro-reflectivity of 142 mcd/(m^2^∙Lux), the LDWS recognition rate decreased to 70% (Table 5). As the precipitation increased to 40 mm/h and 50 mm/h, the lane recognition rate dropped below 50% at a retro-reflectivity of 142 mcd/(m^2^∙Lux). At 104 mcd/(m^2^∙Lux), the recognition rate decreased to 50% when rainfall reached 20 mm/h. Furthermore, with rainfall intensifying beyond 40 mm/h, the rate diminished to below 30%.

### 3.2. Yellow Lane

During the daytime, the yellow lane markings achieved a 100% recognition rate under all rainfall conditions. Even when the retro-reflectivity approached 76 mcd/(m^2^∙Lux), the recognition rate remained above 90% regardless of nighttime lighting or surface moisture. However, during nighttime rain conditions with an intensity of 20 mm/h, the recognition rate fell to 50% at a retro-reflectivity of 76 mcd/(m^2^∙Lux) (Table 6).

### 3.3. Blue Lane

For the blue lane markings, a 100% recognition rate was achieved during daytime conditions. Nighttime trials showed a consistent 90% recognition rate on wet surfaces, independent of lighting presence (Table 7). The recognition rate significantly dropped to 50% at 60 mcd/(m^2^∙Lux) under rainfall intensity of 20 mm/h. Additionally, to maintain a recognition rate above 70% during nighttime rainfall at the same intensity, a retro-reflectivity of at least 130 mcd/(m^2^∙Lux) was required.

### 3.4. Simulation

The simulation framework’s cost analysis, applied to two alternative paint products, A_2_ and A_3_, in addition to the currently used product A_1_, yielded significant insights into efficient road management strategies. The alternatives were defined as A_2_ being more expensive but of higher quality, and A_3_ as a cheaper but lower quality option.

The simulation framework revealed that the product A_3_ emerged as the most cost-efficient option for target recognition rates below 79%. Conversely, the current product A_1_ was found to be the most effective solution for target recognition rates exceeding 79% under the condition of 40 mm/h rainfall intensity and no additional lighting (Figure 6). Additionally, the cost-efficiency ranking of the three alternatives varied depending on different luminance and weather conditions. At a 60% target recognition rate, A_3_ was the most cost-effective product across all conditions. Shifting to a 70% recognition rate, A_1_ outperformed under extreme conditions of heavy rainfall with lighting and moderate rainfall without lighting, while A_3_ remained the preferred option elsewhere. At an 80% recognition target, the trend continued, with A_1_ leading in heavy rain scenarios, whether illuminated or not, and A_3_ favored in less severe weather conditions (Figure 7).

## 4. Discussion

The study aimed to evaluate the impact of weather conditions and wear on the lane recognition rates for blue, white, and yellow road markings. Conducted at the Yeoncheon SOC Demonstration Research Center, various weather scenarios were simulated, incorporating changes in rainfall, lighting, and diurnal cycles. Wear was artificially induced on the markings to assess degradation effects. Recognition rates were determined through ten iterations per condition. The outcomes included: (1) During daytime, the lane recognition rate was 100% under all weather conditions for all colors. (2) At night under heavy rainfall, white lane markings showed a recognition rate below 50% when the retro-reflectivity dropped to 142 and 104 mcd/(m^2^∙Lux). (3) Yellow markings’ recognition halved at night under 20 mm/h rainfall at 76 mcd/(m^2^∙Lux). (4) Over 70% recognition for blue markings at night required a retro-reflectivity of at least 130 mcd/(m^2^∙Lux) under 20 mm/h rainfall.

For stable autonomous driving, a recognition rate above 70% is essential for the functionality of LDWS [26]. In our findings, rainfall exceeding 20 mm/h at night significantly diminished the recognition rate for LDWS. Our research revealed that under such conditions, the recognition rates for white, yellow, and blue lane markings dropped below the critical 70% threshold. Specifically, when the retro-reflectivity fell to 142 mcd/(m^2^∙Lux) for white and 104 mcd/(m^2^∙Lux) for yellow, the recognition rates dropped to around 50%. For blue markings, the recognition rate sharply decreased to 50% when the retro-reflectivity was at 60 mcd/(m^2^∙Lux) during nighttime rainfall. These findings underscore the need for higher retro-reflectivity standards in road markings to support the reliable operation of LDWS in autonomous vehicles, especially under challenging weather conditions.

In our findings, the retro-reflectivity of lane markings was a crucial factor in maintaining high recognition rates for LDWS. High retro-reflectivity levels consistently resulted in robust recognition rates across white, yellow, and blue lane markings regardless of weather conditions, lighting, or the time of day. Specifically, our study demonstrated that when retro-reflectivity was maintained at or above 253 mcd/(m^2^∙Lux) for white lines, 164 mcd/(m^2^∙Lux) for yellow lines, and 132 mcd/(m^2^∙Lux) for blue lines, the LDWS achieved a 100% recognition rate in all tested conditions. Thus, emphasizing the necessity of setting higher retro-reflectivity for road markings would ensure the effective functioning of autonomous driving systems.

Current guidelines for road markings do not consider the increased retro-reflectivity requirements identified in our study, particularly under challenging conditions like heavy rainfall at night. For instance, while standard retro-reflectivity for road markings is typically set below 100 mcd/(m^2^∙Lux) [27], our study indicates that higher values are necessary to maintain effective LDWS performance: 140 mcd/(m^2^∙Lux) for white lines, 160 mcd/(m^2^∙Lux) for yellow lines, and 130 mcd/(m^2^∙Lux) for blue line markings. This difference would arise because conventional standards are designed for human drivers under a broader range of conditions, whereas autonomous vehicles rely heavily on consistent and clear road markings for navigation, especially in challenging environments. Therefore, our findings suggest a need to a revise these standards to accommodate the specific needs of autonomous vehicle technology.

The simulation framework provided a predictive cost analysis based on the degradation rate of the retro-reflectivity over time and its impact on LDWS recognition rates. The results indicate that the most cost-effective product selection depends on the target recognition rate and specific environmental conditions, which underscores the need for a dynamic road maintenance planning process. Such findings have profound implications for transportation policy and the progression of autonomous vehicle infrastructures. They suggest a shift from traditional road maintenance methods towards adaptive strategies that meet the intricate requirements of autonomous navigation systems. Clearly, updating the standards for road marking retro-reflectivity is not only a matter of regulatory compliance, but also a critical step towards optimizing the financial and functional aspects of road infrastructure management in the era of autonomous driving [28,29,30].

There are limitations to consider, although our findings contribute valuable insights into LDWS performance under varying retro-reflectivity levels and weather conditions. Firstly, the specificity of the LDWS models tested may not include the full diversity of systems in use today. Each LDWS may respond differently to the conditions, and further research would be needed to generalize these results across all types of systems. Additionally, the simulated environmental conditions cannot replicate all the conditions of real-world scenarios. Factors such as the interaction with other vehicles, varying light sources, and unexpected road obstruction were not accounted for and could affect the performance of LDWS in practice. Future studies could aim to involve a broader range of LDWS technologies and more complex environmental simulations to build upon the foundation established by the present study.

## 5. Conclusions

In conclusion, this study illuminates the significant impact of retro-reflectivity and adverse weather on the performance of LDWS, while advocating for revised road marking standards required for autonomous vehicle systems. We identified essential retro-reflectivity thresholds for optimal lane recognition, suggesting the need for updated guidelines and advanced LDWS capable of adapting to diverse environmental conditions. Our simulation framework for road marking management provides strategic insights for selecting products based on their performance across varying scenarios. The simulation results underscore the importance of choosing the right road marking products to balance cost and performance, especially in the evolving context of autonomous vehicle technology. This leads to the recommendation that vehicle manufacturers and road maintenance authorities should consider these insights for enhancing autonomous vehicle navigation and safety.

Future research directions include refining our experimental protocol to encompass more nuanced levels of luminance beyond the basic lighting conditions used in this study. Additionally, incorporating a wider range of environmental variables such as regional differences, vehicle traffic frequency, and annual climatic variations will augment the applicability of our simulation framework to real-world road marking maintenance scenarios, further enriching its practical utility and flexibility.

## Figures and Tables

**Figure 1 sensors-24-02505-f001:**
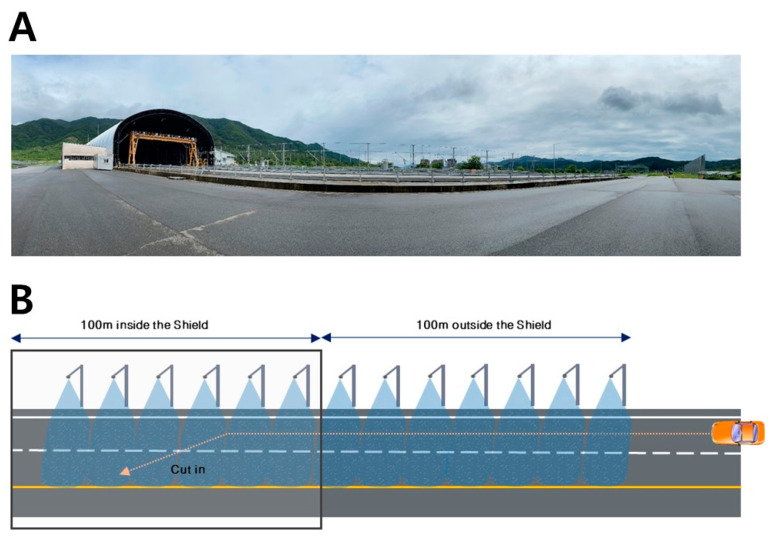
LDWS testing environment and procedure. (**A**) shows the real test track with a shielded section for weather simulation. (**B**) illustrates the test setup where a vehicle equipped with LDWS accelerates to 60 km/h and performs a lane departure at a pre-defined lateral angle. The vehicle starts 100 m outside of the shielded area for acceleration and then enters a 100-m section inside the shield, where the lane departure is executed.

**Figure 2 sensors-24-02505-f002:**
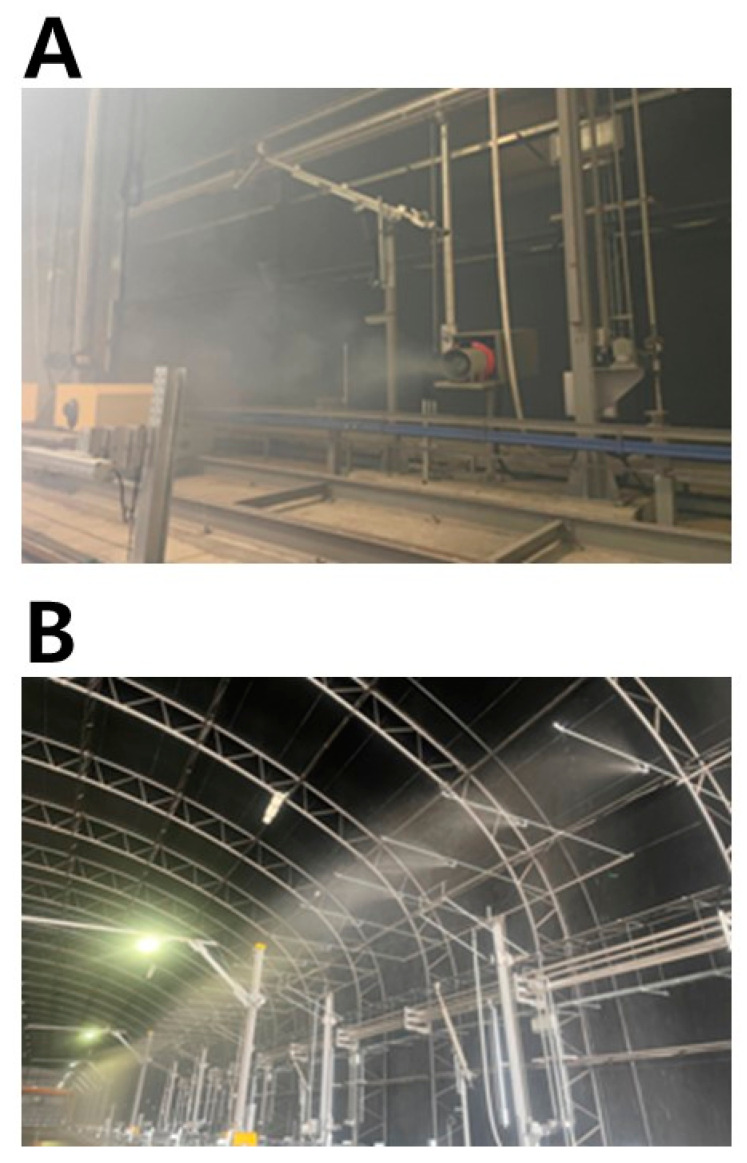
Weather simulation equipment for LDWS testing. (**A**) Fog generation system used to replicate various visibility conditions for assessing LDWS functionality. (**B**) Rain simulation setup designed to create different intensities of rainfall to evaluate the impact on LDWS performance.

**Figure 3 sensors-24-02505-f003:**
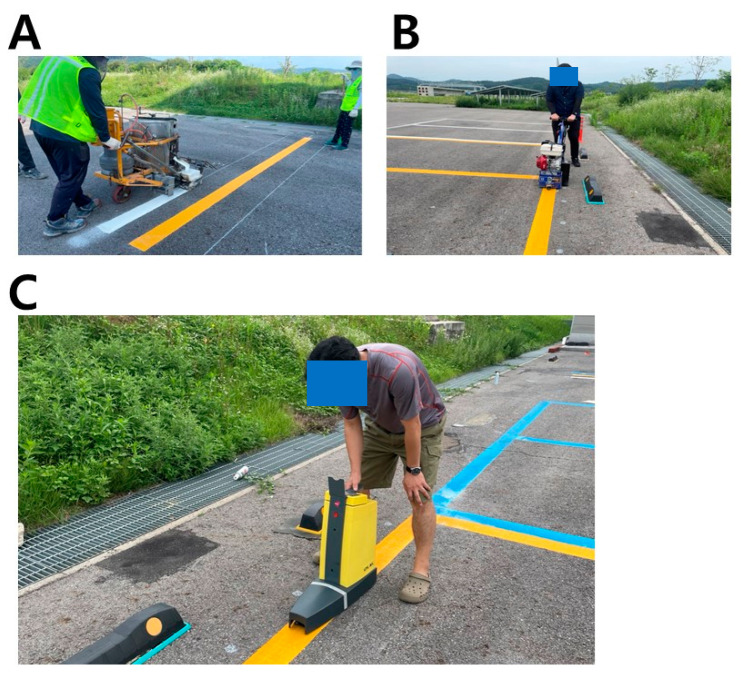
Road marking application (**A**), abrasion process (**B**), and retro-reflectivity measurement (**C**) for preliminary test.

**Figure 4 sensors-24-02505-f004:**
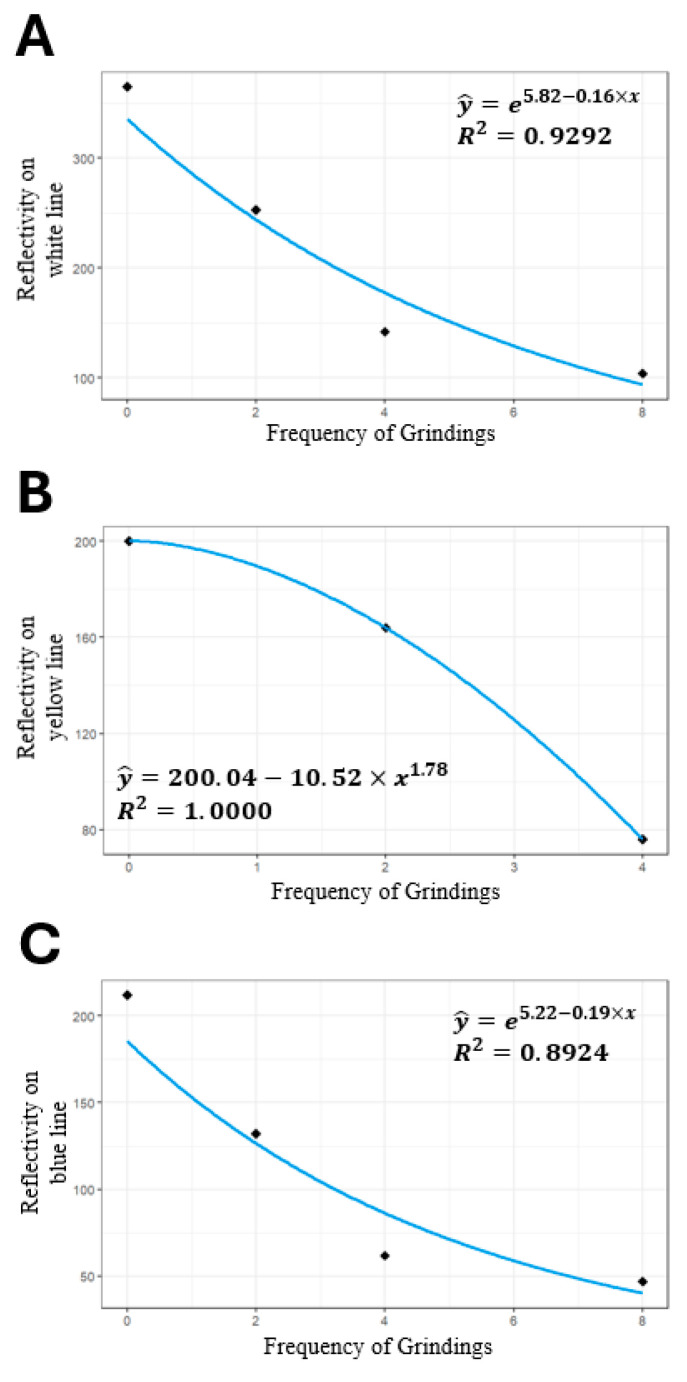
Trend graph of retro-reflectivity coefficients by grinding frequency. This figure illustrates the diminishing retro-reflectivity coefficients of white (**A**), yellow (**B**), and blue (**C**) lane markings as a function of grinding frequency. The *x*-axis represents the abrasion stages, while the *y*-axis measures the retro-reflective coefficients. In all colors, the lane markings exhibit a consistent decline in retro-reflectivity in response to successive grinding stages.

**Figure 5 sensors-24-02505-f005:**
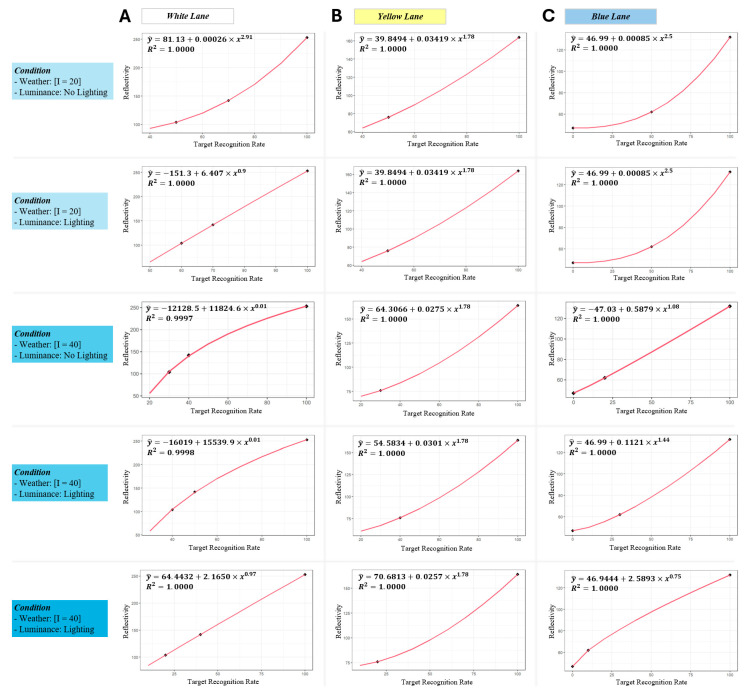
Trend between LDWS recognition rate and retro-reflectivity under the condition of 40 mm/h rainfall intensity and no additional lighting. The red curve line represents the fitted non-linear regression model ($y=\beta_{0}+\beta_{1}\cdot x^{\rho }$), where the hyperparameter ρ is optimized through a grid search in the range of [0, 3] to achieve the maximum *r*^2^ statistics. Note that, as it is not possible to fit a curved line on only 2 observations for yellow lane markings, we assume that the power hyperparameter ρ of yellow lane markings is 1.78, as shown in Figure 4.

**Figure 6 sensors-24-02505-f006:**
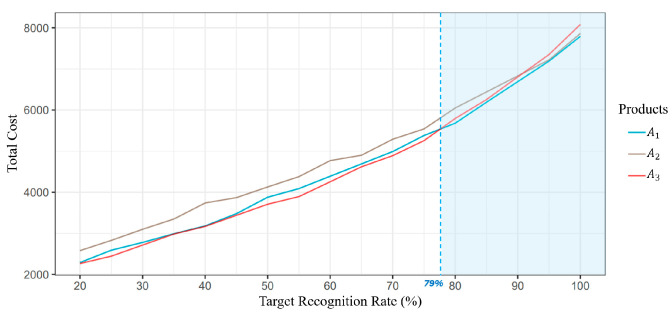
Comparative analysis of total costs for three alternative road marking products ($A_{1},\;A_{2},\;A_{3}$) across a range of target recognition rates. The simulation conditions were set to a rainfall intensity of 40 mm/h and the absence of additional lighting.

**Figure 7 sensors-24-02505-f007:**
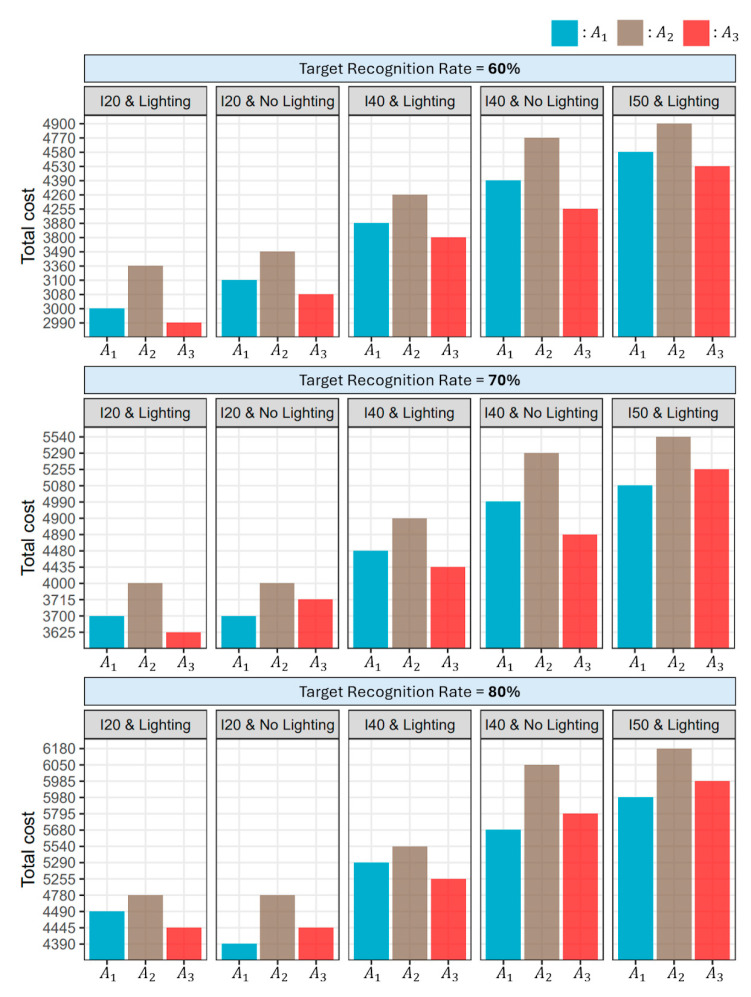
Cost comparison among three alternative road marking products ($A_{1},\;A_{2},\;A_{3}$) at target recognition rates of 60%, 70%, and 80%, under various luminance and weather conditions. The graph illustrates the efficiency of each product, with lower total costs indicating more cost-effective alternatives.

**Table 1 sensors-24-02505-t001:** Retro-reflectivity standards for road markings (manual for installation and management of traffic road markings, Korea National Police Agency, 2020).

Irradiation Angle	Observation Angle	Conditions	Retro-Reflectivity (mcd/(m^2^∙Lux))			
			White	Yellow	Blue	
88.76° (1.24°)	1.05° (2.29°)	Installation	240	150	80	Standard
		Repainting	100	70	40	Recommend
		Rainfall (wet)	100	70	40	Recommend

“Installation” refers to the period from one week after road marking installation up to the completion of the project. “Repainting” is considered when the reflective performance value falls below the standard. The use of products that exceed the Korea Industrial Standards (KSM 6080) is considered a principle. The measurement of the retro-reflective performance of road markings on wet surfaces shall be in accordance with the methods specified in European Standards (EN 1436).

**Table 2 sensors-24-02505-t002:** Evaluation parameters for LDWS performance testing.

Parameters	Conditions
Road marking color	White, yellow, and blue
Road marking wearing(grinding frequency)	No grinding, 2 grindings, 4 grindings, and 8 grindings
Luminance	Daytime, nighttime (road lighting on and vehicle lighting on), and nighttime (road lighting off and vehicle lighting on)
Weather conditions	Normal (dry), wet, rainfall (20 mm/h), rainfall (40 mm/h), fog low visibility (below 50 m), and fog high visibility (below 100 m)

**Table 3 sensors-24-02505-t003:** Measured retro-reflectivity coefficients as a function of grinding frequency for the experiment (unit: mcd/(m^2^∙LUX)).

Color	Initial	2 Grindings	4 Grindings	8 Grindings
White	365	253	142	104
Yellow	200	164	76	-
Blue	212	132	62	47

**Table 4 sensors-24-02505-t004:** Elements of three paint products (unit: 1000 KRW/km for C; mcd/(m^2^∙Lux) for W; days for D).

Product	Elements	Colors		
		White	Yellow	Blue
$A_{1}$ (Our)	Repainting cost (C)	100	110	90
	Retro-reflectivity at installation (W)	365	200	212
	Duration per grind extent (D)	100	120	80
$A_{2}$	Repainting cost (C)	130	140	120
	Retro-reflectivity at installation (W)	380	215	220
	Duration per grind extent (D)	105	130	85
$A_{3}$	Repainting cost (C)	90	95	85
	Retro-reflectivity at installation (W)	360	195	205
	Duration per grind extent (D)	95	100	95

**Table 5 sensors-24-02505-t005:** LDWS recognition rate for white road markings during nighttime (average of 10 trials, unit: %).

	365 mcd/(m^2^ Lux)	253 mcd/(m^2^ Lux)	142 mcd/(m^2^ Lux)	104 mcd/(m^2^ Lux)
Dry and Lighting	100	100	100	100
Dry and No Lighting	100	100	100	100
Wet and Lighting	100	100	90	90
Wet and No Lighting	100	100	90	90
I = 20 (mm/h) and Lighting	100	100	70	60
I = 20 (mm/h) and No Lighting	100	100	70	50
I = 40 (mm/h) and Lighting	100	100	50	40
I = 40 (mm/h) and No Lighting	100	100	40	30
I = 50 (mm/h) and Lighting	100	100	40	20

I = rainfall intensity.

**Table 6 sensors-24-02505-t006:** LDWS recognition rate for yellow road markings during nighttime (average of 10 trials, unit: %).

	200 mcd/(m^2^ Lux)	164 mcd/(m^2^ Lux)	76 mcd/(m^2^ Lux)
Dry and Lighting	100	100	100
Dry and No Lighting	100	100	100
Wet and Lighting	100	100	90
Wet and No Lighting	100	100	90
I = 20 (mm/h) and Lighting	100	100	50
I = 20 (mm/h) and No Lighting	100	100	50
I = 40 (mm/h) and Lighting	100	100	40
I = 40 (mm/h) and No Lighting	100	100	30
I = 50 (mm/h) and Lighting	100	100	20

I = rainfall intensity.

**Table 7 sensors-24-02505-t007:** LDWS recognition rate for blue road markings during nighttime (average of 10 trials, unit: %).

	212 mcd/(m^2^ Lux)	132 mcd/(m^2^ Lux)	62 mcd/(m^2^ Lux)	47 mcd/(m^2^ Lux)
Dry and Lighting	100	100	100	100
Dry and No Lighting	100	100	100	100
Wet and Lighting	100	100	90	90
Wet and No Lighting	100	100	90	90
I = 20 (mm/h) and Lighting	100	100	50	0
I = 20 (mm/h) and No Lighting	100	100	50	0
I = 40 (mm/h) and Lighting	100	100	30	0
I = 40 (mm/h) and No Lighting	100	100	20	0
I = 50 (mm/h) and Lighting	100	100	10	0

I = rainfall intensity.

## Data Availability

The data presented in this study are available upon request from the corresponding author.

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
