# Peer review of "Optimizing Lane Departure Warning System towards AI-Centered Autonomous Vehicles"

_sensors, 2024, doi:10.3390/s24082505_

Round 1

Reviewer 1 Report

Comments and Suggestions for Authors

Please, take care about your abstract and its structure. One common way to structure your abstract is to use the IMRaD structure. This stands for:  Introduction, Methods, Results and Discussion.

Your manuscript needs introduction before the brief literature review. By the way, develop your literature review and finalize it with research gap that you plan fullfil with your results.

Take care on resolution and quality of Fig. 4.

Results on 2, 3 measures are extremely risky. Please, develop Fig. 5. It reminds a theoretical rather than comming from practise.

Conclusion is weak, too short. It does not contain future research and research limitations.

Comments on the Quality of English Language

Minor editing of English language are required.

Reviewer 2 Report

Comments and Suggestions for Authors

In this paper, the influence of road marking reflectivity on LDWS performance is studied. The recognition rate of road markings under different environmental conditions, including different weather conditions and light changes, was investigated. The performance of LDWS was evaluated by applying controlled wear to white, yellow and blue road markings and measuring their reflective properties at different stages of degradation. The experimental method is reasonable to some extent, and the simulation framework is proposed to adapt to some scenarios. The following are some suggestions:

1. In introduction, it is suggested to add the method summary of similar studies to highlight the innovation of this paper.

2. In section “Materials and Methods”, it is recommended to describe the basic models and parameters of different experimental groups of vehicles and their lane keeping systems.

3. It is suggested to show the overall data pattern to clarify the rationality of adopting this fitting method.

4. In subsection 2.6 Experimental procedures, it should be described about the definition of the search scope of the hyperparameter ρ.

5. It is suggested to clarify the calculation method of target recognition rates for main evaluation indicators. Is it the same accuracy as in machine vision.

6. The accuracy rate of vision-based assisted driving is greatly affected by the overall tone of the environment. In the different experiments of different color lanes in this paper, is the influence of the environment on the variable of lane recognition considered?

Round 2

Reviewer 2 Report

Comments and Suggestions for Authors

The authors have answered all my questions and addressed all my concerns in the revision. The revised manuscript is improved in many aspects. I think this paper can be accepted for publication.